# Fantastic Tractor-Dogs and How Not to Find Them With Open-Vocabulary Detectors

**Frank Ruis, Gertjan J. Burghouts, & Hugo Kuijf**
TNO, Intelligent Imaging
`frank.ruis@tno.nl`

## Abstract

Open-Vocabulary Detectors (OVDs) excel in zero-shot benchmarks, but we observe a critical flaw in real-world deployment: a high rate of confident false positive predictions on images that do not contain any target objects (e.g., detecting a tractor in an image of a dog). This issue is masked by standard benchmarks like COCO and LVIS, as they rarely contain images without any of the target classes present. We identify vision-language fusion layers in early-fusion OVD architectures (e.g., Grounding DINO or LLMDet) as the root cause, and show how they distribute irrelevant class information across image features when no prompted object is present. To mitigate background false positives without costly retraining, we propose a simple, training-free method: appending attention sink tokens to the input prompt. We show that such sinks can redirect spurious attention and dramatically reduce background false positives. Our approach significantly improves the performance of all six early-fusion models tested (e.g., boosting AP on LVIS by more than 5x at a false positive rate of 0.01 for some models), making them practical for real-world applications where images without the object of interest are much more prevalent.

## 1 Introduction

Open-Vocabulary Detection (OVD) models boast increasingly impressive performance on zero-shot benchmarks (Liu et al., 2024b; Ren et al., 2024b;a; Minderer et al., 2023), but their propensity for false positive predictions severely limits their applicability outside academic contexts. An example of this is shown in Figure 1, where any prompted class or phrase will result in confident false positive predictions on semantically irrelevant objects. While the literature on Vision-Language Model (VLM) hallucination (Sarkar et al., 2025; Yang et al., 2025; Liu et al., 2025; Min et al., 2025) is steadily growing, there is a notable lack of similar studies focused on OVD models. Although the issue has been noticed for quite some time by practitioners (GitHub Issue, 2023), it remains largely unexplored in academia, apart from a brief mention in the Grounding DINO 1.5 Technical Report (Ren et al., 2024b), highlighting the increased use of negative samples during pre-training. One possible reason for this gap is that false positives in OVD models tend to occur primarily in background-only images, where none of the target classes are present. However, standard training datasets and benchmarks such as COCO (Lin et al., 2015) and LVIS (Gupta et al., 2019) almost always include at least one annotated ground truth instance per image. This leads to strong benchmark performance, yet fails to reflect many real-world scenarios where background images are far more common than those containing foreground instances, such as in security or medical imaging.

We investigate the cause of this background false positive rate ($FPR_{bg}$) and highlight the differences between early-fusion and late-interaction OVD models with respect to $FPR_{bg}$. Specifically, we quantitatively show that only early-fusion models suffer from a high $FPR_{bg}$ compared to their FPR on foreground images. For late-interaction models, there is no significant deviation between the two. Because early fusion models offer strong benefits over late interaction models in certain complex tasks, such as visual question answering (Marino et al., 2019; Goyal et al., 2017) and referring expression comprehension (Kazemzadeh et al., 2014; He et al., 2023), it would be beneficial to mitigate their high $FPR_{bg}$. We establish that cross-modal attention operations in the vision-language fusion layers of early-fusion models are the cause of the high $FPR_{bg}$, and propose a training-free attention sink approach to almost completely mitigate the issue. This finding not only reveals a

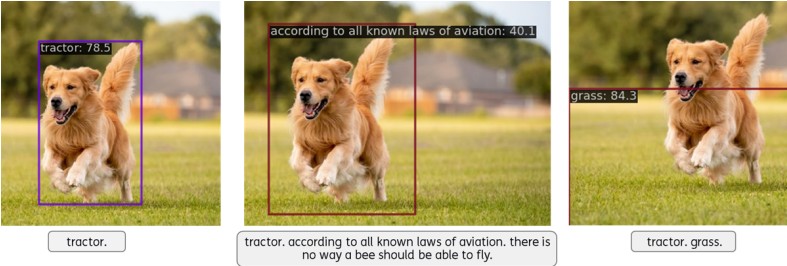

Figure 1: Example of Grounding DINO (Liu et al., 2024b) confidently predicting "tractor" on a picture of a golden retriever. We show that this is a common error for open-vocabulary detectors when prompting for an object that does not occur in the image.

previously overlooked limitation in the architectural design of early-fusion OVD models, but also opens up a broader line of inquiry into how cross-modal attention mechanisms can be rethought to balance generalisation with reliability in real-world deployments.

Our main contributions can be summarised as follows:

- We identify a pervasive issue of confident false positive predictions across 4 state-of-the-art open-vocabulary object detectors, significantly impacting their practical usability, and show why this has not been identified by any commonly used object detection benchmarks.

- We propose a simple adaptation to existing benchmarks to identify and quantify background-only false positives.

- We show that late-interaction models do not behave differently on background images, while early-fusion models do, and offer a potential explanation of its cause through inspecting the attention patterns in vision-language fusion layers.

- We propose a training-free approach to mitigate background-only false positives: simply appending attention sink tokens to the prompt to reroute attention away from negative classes. This results in significant improvements in false-positive mitigation with minimal impact on the performance on positive samples.

## 2 PRELIMINARY: EARLY FUSION AND LATE INTERACTION

OVD models can be broadly categorised as early fusion or late interaction, depending on how and at which point in the architecture the vision and language modalities interact (cf. Figure 2). As the name implies, early fusion models, such as GLIP (Li et al., 2022) and Grounding DINO (Liu et al., 2024b), combine vision and language features, usually through a cross-attention operation, followed by further processing by additional modules. Late interaction models, such as CLIP (Radford et al., 2021) or OWL-ViT (Minderer et al., 2022), never fuse vision and language features but align the feature outputs of two modality-specific encoders through a vector similarity metric.

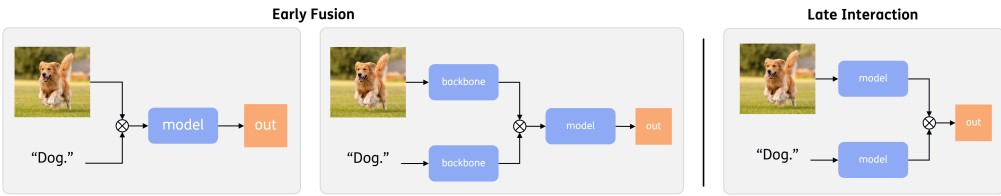

Figure 2: In early fusion models the vision and language features are combined, usually through a cross-attention operation, followed by further processing by additional modules. Late interaction models never fuse features, but align the feature outputs of two modality-specific encoders through a vector similarity metric.

Although late interaction performs well for tasks such as retrieval and regular object detection (Cheng et al., 2024; Minderer et al., 2023), the lack of cross-modal interaction means it cannot effectively perform more complex tasks such as referring expression comprehension (REC) or visual question answering (VQA) (Shin et al., 2022; Alberti et al., 2019). The current state-of-the-art (Ren et al., 2024a) suggests that early fusion methods also outperform late interaction methods in tasks such as regular object detection; however, in Section 3 we show that this is not necessarily the case.

## 3    IDENTIFYING THE FALSE POSITIVE PROBLEM

**Quantification**    As mentioned in the Introduction 1, standard benchmarks such as COCO (Lin et al., 2015) and LVIS (Gupta et al., 2019) mask the issue of background-only false positives because they rarely contain background images. Even if they do, common training configurations in popular frameworks such as mmdetection (Chen et al., 2019) are set up to filter out images that do not contain ground truth boxes. LVIS is a federated dataset, which means that its classes are not exhaustively labelled for every image. Instead, each individual image has a set of positive classes that have been labelled, as well as a set of negative classes that have been confirmed to not be visible in the image. We leverage these negative labels to quantify the problem; by prompting the model with exclusively negative classes, we know that any prediction above a given confidence threshold can be considered a false positive.

For each image, we loop over all positive and negative labels and prompt the OVD model one class at a time. The predictions on positive labels are used to calculate the average precision (AP) as usual. The predictions of the negative labels are used to determine the number of false positives per background image, which we denote as $\text{FPR}_{\text{bg}}$. The $\text{FPR}_{\text{bg}}$ can be decreased by increasing the confidence threshold for accepted predictions. We can calculate an $\text{AP}^{\text{FPR}}_{\text{fpr}}$ at a specific $\text{FPR}_{\text{bg}}$ value of $fpr$ by increasing the confidence threshold for predictions until the desired $\text{FPR}_{\text{bg}}$ value is reached. Once the desired $\text{FPR}_{\text{bg}}$ is reached, we filter out positive predictions below the new confidence threshold and calculate the corresponding AP.

**Early fusion models degrade on background images**    Figure 3 shows the LVIS MiniVal AP of eight OVD models compared to their $\text{AP}^{\text{FPR}}_{0.05}$. Ideally, models that do not behave differently on background-only images should lie along the diagonal, indicating no drop in performance. The vertical deviation below the diagonal signifies the model's propensity towards background-only false positives. We can see a clear divide between late-interaction and early-fusion models: Late-interaction models behave similarly to regular (non-OVD) models in that they may have some false positives on background images, but not to the extent that it significantly impacts their performance on positive samples when picking a confidence threshold. Early fusion models, on the other hand, experience a significant drop in performance on positive samples when we lower their $\text{FPR}_{\text{bg}}$.

**Why not use only late interaction architectures?**    A simple solution then seems to be to just not use early fusion, which may be valid in certain use-cases. However, early-fusion models offer several key advantages over late-interaction models, such as improved compositional and out-of-domain generalisation (Zhou et al., 2022), and the ability to perform more complex tasks such as Referring Expression Comprehension (REC) or Visual Question Answering (VQA), which are ill-suited for late-interaction architectures (Shin et al., 2022; Alberti et al., 2019).

**Earlier solution attempts are not scalable**    To increase robustness to $\text{FPR}_{\text{bg}}$, the authors of Grounding DINO, for their subsequent closed source models, sample significant amounts of additional negative examples during pre-training, as mentioned in the Grounding DINO 1.5 technical report (Ren et al., 2024b). However, this approach does not seem scalable, given that it remains exceedingly easy to find irrelevant classes with confident false positive predictions even for their most potent models (e.g. a red panda being detected as jenga blocks, cf. Figure 10 in the Appendix). It may be hard or even impossible to train on sufficient negative examples, due to the combinatorial explosion of possible negative classes and objects in images.

**There is no "None of the Above"**    Clearly, early-fusion models are capable of recognising that a dog is in fact not a tractor, as evidenced by the fact that the mistake usually disappears when adding

**False Positive Mitigation with Attention Sinks**

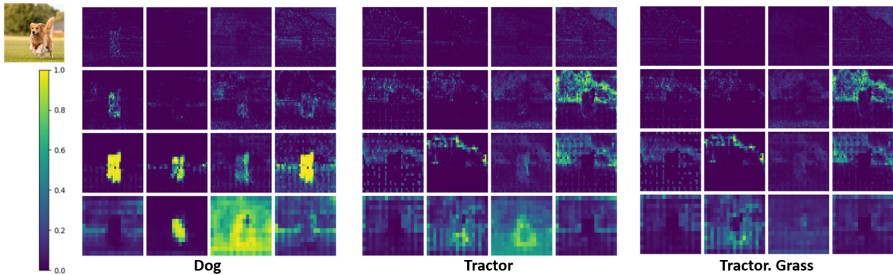

```python
from mmdet.apis import DetInferencer

inferencer = DetInferencer(model='cfg_mmdet.py'
                           , weights='base.pth')
nsi = 24
si = [f"[unused{i}]" for i in range(nsi)]
inferencer.model.language_model.tokenizer.add_tokens(si)

# E := inferencer.model ... word_embeddings.weight.data
E[1:nsi+1] = E.mean(dim=0).unsqueeze(0).repeat(nsi, 1)

cls = ["tractor"]
rs = inferencer("dog.jpg", show=True, texts=cls + si)

# filter out predictions using sinks
keep = [e < len(cls) for e in rs['predictions'][0]['labels']]
rs = {k: v for i, (k, v) in enumerate(rs.items()) if keep[i]}
```

Figure 3: On the left, we show the LVIS MiniVal AP compared to the AP at an $FPR_{bg}$ of $0.05$ for 8 open-vocabulary detectors, in a zero-shot setting. While most models perform worse when taking background images into account (below the diagonal), the performance drop is especially severe for all early fusion models (lower right). Adding attention sinks ($\star$) significantly improves performance for all tested models. On the right, a stand-alone snippet of code showing the ease of implementing attention sinks for MM-Grounding DINO.

a different positive class to the prompt. Inspecting the attention patterns of vision-language fusion layers, we find that the issue likely stems from the nature of the attention operation not being able to pick zero tokens in the absence of a good match. Figure 4 shows an example of such a visualisation for a positive, negative, and positive + negative prompt. For the negative prompt *tractor*, every vision token incorporates at least some information of the tractor class since there is no other prompt token to choose from. However, when adding the positive class *grass*, the vision tokens containing grass can instead attend to the grass class, while the other vision tokens can incorporate a mix of both classes. This seems to be enough for the model to calibrate its confidence so that only grass will be detected. Adding one or more additional negative classes to the prompt does not mitigate the number of false positives for background-only images, the most "prevalent" class features in the vision tokens will always get a high confidence score. We hypothesise that in a later stage the model calibrates confidence scores according to the relative proportions of class information incorporated by each of the vision tokens, such as in the grass example. We observe that, in the absence of a clear positive class, one of the negative classes will be picked instead, leading to confident false positives.

Figure 4: A visualisation of the attention scores of each head (horizontal) and scale (vertical) from the first vision-language fusion layer of LLMDet (Fu et al., 2025), between the visual features of an image of a golden retriever and the prompt tokens "dog" (left) and "tractor" (middle and right). When prompted for a tractor (middle), significant attention is routed to the visual features of a dog, causing the classification head to predict the tractor class when presented with the class-agnostic object query containing the dog features mixed with the tractor token. Adding a positive class (grass, right) partly mitigates that, due to the model being better at calibrating confidence scores when a positive class is present.

## 4   REDISTRIBUTING IRRELEVANT INFORMATION TO ATTENTION SINKS

Recent work in both language and vision has shown that attention sinks (Xiao et al., 2023), or register tokens (Darcet et al., 2023), may be needed to prevent transformer-based models from reusing certain input tokens as scratch pads for unrelated computations. In practice, these models often re-purpose low-information tokens to store global information. While this mechanism is helpful for achieving strong overall performance, it disrupts locality of information and can hurt performance on dense prediction tasks (Darcet et al., 2023). Interestingly, this behaviour differs from what we observe in the vision-language fusion layers of OVD models (Section 3). Instead of concentrating important information into a few tokens, these models spread irrelevant class information across many tokens.

This diffusion of irrelevant information is consistent with how such models are trained. Since every image always contains at least one positive class, and irrelevant information cannot simply be discarded, the model only needs to dilute irrelevant signals enough so that they remain below the confidence threshold of the positive class in the prediction heads. However, when no positive class is present, this irrelevant information becomes the dominant class signal in the features. Luckily, this does suggest that the model has already learned to minimize the presence of irrelevant information in its features. To prevent false positives, we only need to allow the model to pick features that do not pertain to any class in particular.

Here is where attention sinks can serve a useful role, specifically, as the missing "none of the above" option discussed in the previous section. By introducing one or more semantically neutral sink tokens into the input prompt and treating them as regular target classes, the model naturally routes excess or irrelevant attention to these sinks. When the model assigns a sink token as the predicted class for an object, that prediction can safely be discarded (cf. Figure 5). This approach reduces background false positives while preserving the model's calibration: positive class predictions remain largely unaffected even after sink tokens are integrated into the feature map.

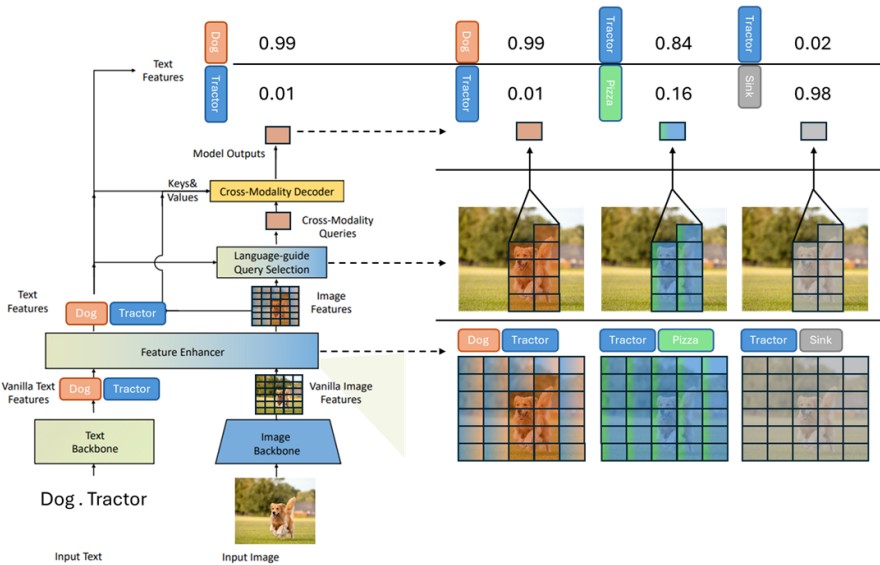

Figure 5: A diagram showing an example of attention patterns for queries containing a positive, only negative, and negative and attention sink tokens respectively. Modification of an image taken from Grounding DINO (Liu et al., 2024b).

To initialise the attention sink tokens $\mathbf{s}_i$, we use one of three initialisation strategies:

$$
\mathbf{s}_i \in
\begin{cases}
\mathcal{N}(0, \sigma^2 I) & \text{(random init)} \\
\frac{1}{|V|} \sum_{w \in V} \mathbf{e}_w & \text{(mean of all word embeddings } \mathbf{e}_w \text{ in vocabulary } V) \\
\mathbf{e}_{[()]} & \text{(embedding of special-character string "[()]" )}
\end{cases}
\quad \text{for } i = 1, \ldots, N_{\text{sinks}},
$$

We find that different initialisations work better for different models and elaborate on our choices in Appendix A.2. The attention sinks work out-of-the-box with all models evaluated in the paper, without the need for any training.

In addition to language sinks in the prompt, we also experimented with vision sinks by introducing extra sink tokens into the vision features at every language-vision fusion step. While this alone can improve performance, we observed that false positives are primarily driven by the vision-to-language attention direction. With well-chosen language sink initialisations, the need for vision sinks largely disappears. This suggests that the effect of irrelevant label information in vision tokens is minimal, likely due to the large number of vision tokens available, and their existing integration of language sink features when also using language sinks.

## 5 EXPERIMENTAL SETTINGS

We study how well attention sinks can mitigate background false positives ($FPR_{bg}$) on common OVD benchmarks, ideally without affecting the performance on positive classes. Attention sinks can be implemented in just a couple of lines of code, an example of which can be seen in Appendix A.1.

**Benchmarks** The POPE benchmark (Li et al., 2023) probes large vision-language models for object hallucinations. It uses images and annotations from COCO (Lin et al., 2015), with binary labels that denote the presence of a given object. These are used to evaluate vision-language models with language outputs on yes-no questions. We convert POPE back to the object detection task by using the ground-truth bounding boxes of a given POPE object and image instead of a binary label, and use the standard average precision (AP) metric over model predictions. For negative classes, we record all predictions and their confidence scores to determine the $FPR_{bg}$ at a given confidence threshold. An added benefit of POPE's approach of probing for one object at a time is that this eliminates the large variance in prediction results for early-fusion models when changing the order or number of classes in the prompt.

The experiments for other datasets are constructed analogously to POPE. LVIS (Gupta et al., 2019) is the main benchmark used to evaluate the zero-shot generalisation for open-vocabulary object detectors. Because LVIS is a federated dataset, we use the explicit negative classes provided for each image instead. The goal is to reach a high AP while keeping $FPR_{bg}$ low. The FPR is defined as the number of false positives per *prompt*, i.e. an FPR of 0.75 on LVIS indicates that you can expect any of the 1203 LVIS classes to result in a false positive prediction on 75% of images that do not contain that class.

In the following sections, we focus on the LVIS dataset because it is the most comprehensive and widely used dataset for open-vocabulary detection. Results of additional experiments, including POPE, can be found in Appendix A.3.

**Models** We evaluate five early fusion models, GLIP (Li et al., 2022), OmDet-Turbo (Zhao et al., 2024a), Grounding DINO (Liu et al., 2024b), MM-Grounding DINO (Zhao et al., 2024b), and LLMDet (Fu et al., 2025), and three late interaction models, YOLO-World (Cheng et al., 2024), OV-DINO (Wang et al., 2024b), and OWL-ViTv2 (Minderer et al., 2023).

**Attention Sinks** We explore the optimal number of attention sinks for various models as well as the type of initialisation of the attention sinks (random, mean, specific words or tokens). We also explored learning the word embedding weights of the attention sinks, which we elaborate on in Appendix A.4, but that did not offer improvements over the training-free approach. Ablations are performed on 64 images of the VOC dataset, with two positive and two negative prompts per image. We combined the attention sink initialisation experiment with a search of the minimum number of images from the 22-class VOC dataset that can offer a stable performance indicator of the more complex 1203-class LVIS dataset. We did so by first searching over a range of 8, 16, 32, 64, 128, 256, 512, and 1024 images to find the minimum number of VOC images that maintains a stable performance over different subsets of images and then compared that performance to the same attention sink initialisation on the LVIS MiniVal dataset.

## 6 ANALYSES AND FINDINGS

**Can attention sinks reduce background false positives?** In Figure 3, we showed that early fusion models experience a sharp performance drop on LVIS once background false positives are considered. For example, on LVIS MiniVal, LLMDet Swin-B achieves an AP of 0.466. However, this comes with an $FPR_{bg}$ of 2.4 at a confidence threshold of 0.25, i.e., more than two false positives per image and negative prompt on average.

Table 1 further details the LVIS MiniVal results of eight OVD models at five different $FPR_{bg}$ values, without and with our attention sinks. Attention sinks aim to keep model performance at low $FPR_{bg}$ close to the standard AP (final column), which would indicate full mitigation of background false positives without hurting positive-class detection. The bottom section of Table 1 shows early-fusion models augmented with attention sinks, with absolute improvements highlighted in green. At low $FPR_{bg}$, all models see large AP gains. While late-interaction models still achieve the best performance at near-zero false-positive rates, attention sinks make early-fusion models competitive at practical error levels. For example, LLMDet-T more than doubles its AP at $FPR_{bg} = 0.05$ and achieves a five-fold increase at $FPR_{bg} = 0.01$. Some models show slight drops in AP when ignoring background false positives (final column), largely due to low-confidence predictions falling below the threshold after attention sinks are applied (see Figure 6 for an example). Crucially, almost all background false positives disappear. The few that remain are similar to the errors seen in late-interaction models, such as fine-grained errors (e.g., a Labrador detected as a Cocker Spaniel) or visual similarity errors (e.g., a regular pumpkin detected as a jack-o'-lantern).

| Model | $AP^{FPR}_{0.01}$ | $AP^{FPR}_{0.05}$ | $AP^{FPR}_{0.1}$ | $AP^{FPR}_{0.25}$ | AP |
|---|---|---|---|---|---|
| *Early-Fusion* | | | | | |
| OmDet Turbo | 0.035 | 0.075 | 0.093 | 0.127 | 0.228 |
| GLIP Swin-T | 0.116 | 0.180 | 0.221 | 0.257 | 0.333 |
| GDINO Swin-T | 0.037 | 0.098 | 0.141 | 0.211 | 0.396 |
| MM-GDINO Swin-T | 0.076 | 0.148 | 0.184 | 0.257 | 0.407 |
| LLMDet Swin-T | 0.045 | 0.140 | 0.195 | 0.258 | 0.464 |
| LLMDet Swin-B | 0.047 | 0.125 | 0.166 | 0.218 | 0.466 |
| LLMDet Swin-L | 0.113 | 0.218 | 0.263 | 0.340 | 0.488 |
| *Late-Interaction* | | | | | |
| YOLO-World L | 0.189 | 0.238 | 0.245 | 0.245 | 0.245 |
| OV-DINO Swin-T | 0.274 | 0.316 | 0.325 | 0.336 | 0.349 |
| *Early-Fusion + attn sinks* | | | | | |
| GLIP Swin-T | 0.166 (+0.050) | 0.229 (+0.049) | 0.256 (+0.035) | 0.283 (+0.026) | 0.302 (-0.031) |
| GDINO Swin-T | 0.073 (+0.036) | 0.152 (+0.054) | 0.192 (+0.051) | 0.256 (+0.045) | 0.359 (-0.037) |
| MM-GDINO Swin-T | 0.214 (+0.138) | 0.306 (+0.158) | 0.348 (+0.164) | 0.394 (+0.137) | 0.426 (+0.019) |
| LLMDet Swin-T | 0.223 (+0.178) | 0.326 (+0.186) | 0.359 (+0.164) | 0.400 (+0.142) | 0.448 (-0.016) |
| LLMDet Swin-B | 0.222 (+0.175) | 0.349 (+0.224) | 0.403 (+0.237) | 0.449 (+0.231) | 0.499 (+0.033) |
| LLMDet Swin-L | 0.182 (+0.069) | 0.318 (+0.100) | 0.387 (+0.124) | 0.442 (+0.102) | 0.502 (+0.014) |

Table 1: Results on our proposed LVIS hallucination detection benchmark, at five levels of $FPR_{bg}$. Although late-interaction models still outperform early-fusion models at near-zero false positive rates, attention sinks make early-fusion models competitive again at manageable false positive rates.

**What is the effect of attention sinks on the vision-language fusion layers?** Figure 7 visualises the attention scores from the first vision-language fusion layer of LLMDet-T. Without attention sinks, almost all visual features attend to the negative class token *tractor*, which confuses the classification head. Adding attention sinks redirects this spurious attention to sink tokens, leaving the tractor token with the desired near-zero attention. For the true positive class *dog*, the attention map also becomes much cleaner: background tokens no longer absorb information from the dog token. As a result, the classification head avoids the calibration issues seen without attention sinks.

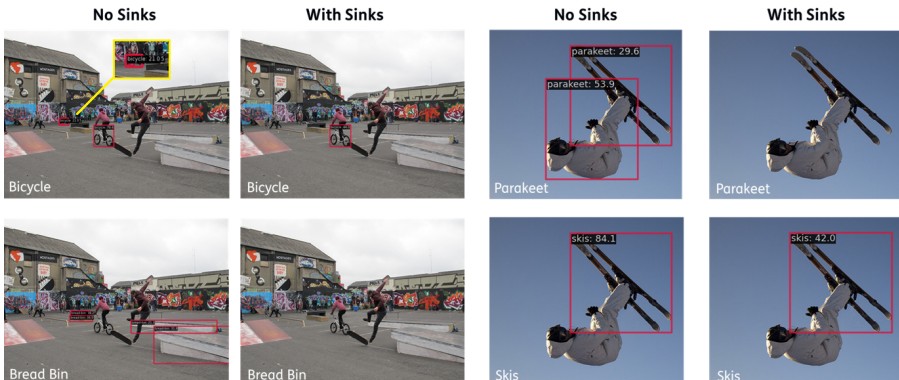

Figure 6: Qualitative examples of zero-shot predictions with LLMDet Swin-T on LVIS MiniVal images, with and without using attention sinks. Without attention sinks, the model has many false positive predictions regardless of semantic relevance, e.g. misidentifying skis as a parakeet. While in some cases low-confidence true positives will become false negatives when using attention sinks (e.g. one of the bicycles), almost all cases of background false positives are mitigated.

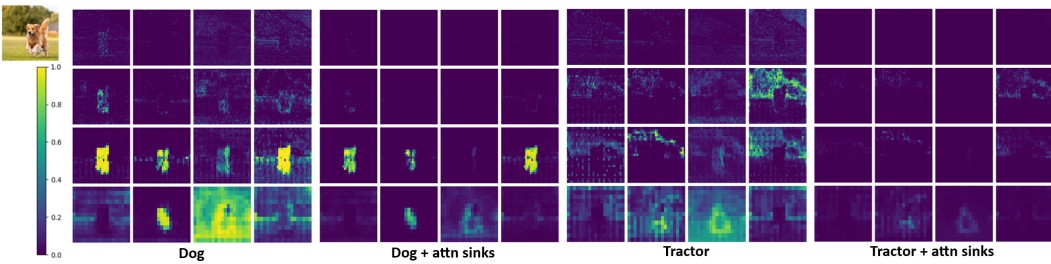

Figure 7: A visualisation of the attention scores of each head (horizontal) and scale (vertical) from the first vision-language fusion layer of LLMDet, between the visual features of an image of a golden retriever and the prompt tokens "dog" (top) and "tractor" (bottom), without using attention sinks (left) and with attention sinks (right). Both the positive and negative class have a much cleaner attention map after adding attention sinks, with most irrelevant information being routed away from the negative classes to the attention sinks.

**How do we find the right attention sinks?** We investigate both the initialisation strategy and the number of attention sinks, with results shown in Figure 8. Using only 64 VOC images is sufficient to predict how an initialisation strategy will perform on the full LVIS MiniVal set, enabling rapid evaluation of many candidates. This is especially important since no single "one-size-fits-all" strategy works across models. LLMDet, in particular, diverges from other early-fusion models, likely due to its LLM-aided training altering vision–language interactions.

Figure 8 (right) shows the effect of the number of sinks for LLMDet-T. Here, 24 attention sinks perform best on LVIS MiniVal, while gains diminish sharply after 8. Across all models, using more than the optimal number never harms detection performance. Multiple sinks can even share the same initial embedding, as positional encodings introduce sufficient variability.

## 7 RELATED WORK

**Open-Vocabulary Detectors** (OVD) were first introduced as an extension of CLIP (Radford et al., 2021) to object detection, as explored by (Gu et al., 2022). Building on this idea, GLIP (Li et al., 2022) took a different approach by training an object detector from scratch rather than adapting a pre-trained CLIP model. Since then, both directions have seen significant advancements, with models such as OWL (Minderer et al., 2022), OWLv2 (Minderer et al., 2023), and the Grounding DINO family (Liu et al., 2024b; Ren et al., 2024b;a). The current state of the art among openly available models is set by OV-DINO (Wang et al., 2024b) and LLMDet (Fu et al., 2025). More

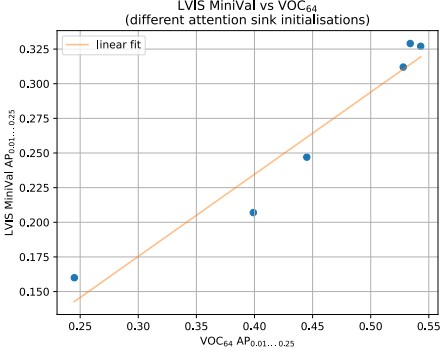 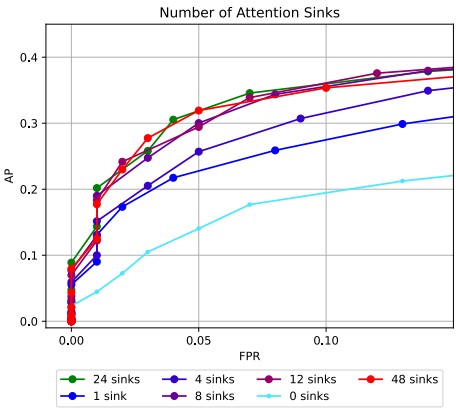

Figure 8: **Left**: Comparing 6 different attention sink initialisation strategies and the corresponding performance on 64 images (x-axis) of the 20 class VOC dataset vs. the full 1203 class LVIS MiniVal dataset (y-axis). There is a strong positive linear correlation, meaning a small number of images of a relatively narrow dataset can accurately predict the performance of attention sinks on a much broader range of classes. **Right**: Exploring the effect of the number of attention sinks on $FPR_{bg}$, using LLMDet-T. Performance gains start diminishing from about 8 sinks, with the optimal number of attention sinks being around 24 tokens.

recently, OVDs have also been streamlined for real-time applications, exemplified by YOLO-World (Cheng et al., 2024), YOLOE (Wang et al., 2025), and OmDet Turbo (Zhao et al., 2024a).

**Vision-Language Hallucination**   Because false positives in OVD models arise due to vision-language fusion operations, there are some similarities with research on hallucination prevention for large vision-language models. Earlier work focused on methods such as instruction tuning (Wang et al., 2024a; Liu et al., 2024a) or post-hoc remedies such as hallucination detection and correction (Gunjal et al., 2024; Yin et al., 2024), but more recent work has identified issues in attention operations which focus mainly on preceding text instead of vision tokens (Sarkar et al., 2025; Yang et al., 2025; Liu et al., 2025; Min et al., 2025).

A similar pattern arises for open-vocabulary detection, where newer Grounding DINO models attempt to mitigate false positives through training on additional negatives (Ren et al., 2024b), practitioners have started applying post-hoc methods such as verification of OVD detections with CLIP models (GitHub Issue, 2023), and in this paper we identify and propose a remedy for issues in vision-language attention operations.

**Attention Sinks**   Concurrent work in computer vision (Darcet et al., 2023) and natural language processing (Xiao et al., 2023) has shown the propensity of large transformer models to repurpose input tokens as a sort of scratch pad for calculations that are not necessarily related to the specific tokens in question. In the case of vision transformers, this may lead to issues when certain assumptions are made, such as feature locality for dense prediction, which does not hold for the scratch pad tokens (Darcet et al., 2023). The introduction of attention sink tokens, learnt or constructed (Jiang et al., 2025), can redirect scratch pad attention to special tokens unrelated to the input image. This keeps feature locality, while retaining the benefits of scratch pad tokens. For Xiao et al. (2023), the problem was the deterioration of long-context language modelling once certain attention sink tokens exit the context window, with the solution to simply always keep those tokens in the context.

Instead of re-routing or keeping track of existing information, we utilise attention sinks as a semantically neutral attention source. This allows the model to pick sink tokens over target classes if none of the target classes is present in the image.

## 8 CONCLUSION

We have identified and quantified the pervasive issue of false positive predictions on background images for open-vocabulary detectors with early-fusion architectures. We identified their vision-language fusion layers as the cause of these errors, specifically the fact that these layers do not have the ability to ignore all target class tokens in the prompt when none are relevant. We proposed training-free attention sinks as a cheap and easy way to mitigate the issue, attention sinks improve AP up to 5x at low false positive rates. Although we were unable to find a one-size-fits-all solution in terms of attention sink initialisation strategy, we showed that candidate strategies can be efficiently evaluated on a small set of images. We encourage future work on training open-vocabulary detectors with early-fusion architectures from scratch to incorporate gated attention or a similar option that explicitly allows the model to discard irrelevant class information.

## 9 REPRODUCIBILITY STATEMENT

All details needed to reproduce our results, including a stand-alone code snippet, can be found in Section 5 and Appendix A.1.

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

## A APPENDIX

### A.1 IMPLEMENTATION

The implementation is very straightforward. All that is needed to implement training-free attention sinks for an MM-Grounding DINO model in mmdetection, for example, is the following:

```python
from mmdet.apis import DetInferencer

inferencer = DetInferencer(model='cfg_llmdet.py', weights='base.pth')

nsi = 24
si = [f"[unused{i}]" for i in range(nsi)]
inferencer.model.language_model.tokenizer.add_tokens(si)

# E := inferencer.model.language_model ... word_embeddings.weight.data
E[1:nsi+1] = E.mean(dim=0).unsqueeze(0).repeat(nsi, 1)

classes = ["tractor"]
result = inferencer("dog.jpg", show=True, texts=classes + si)

# filter out predictions using sinks
keep = [e < len(classes) for e in result['predictions'][0]['labels']]
result = {k: v for i, (k, v) in enumerate(result.items()) if keep[i]}
```

Here $E$ is shorthand for the word embedding weights of the Embeddings module of the language backbone. By appending the *si* array to your prompt, attention sinks will be enabled, and false positives will be mitigated. All that is left to do after inference is to discard any predictions with one of the attention sinks as the predicted class.

## A.2  CHOICE OF INITIALISATION STRATEGY

Intuitively, a good training-free attention sink should have a high enough activation with any vision token to overrule irrelevant classes in the prompt, but not too high as to impact a positive class. Some options we evaluated are as follows:

- **Random**: During training, the vision backbone should have learnt to increase the embedding distance to irrelevant classes and lower the distance to relevant classes. This may in turn result in completely random embeddings ending up somewhere in-between. We find that this initialisation strategy performs best for the original Grounding DINO models.

- **Mean**: The mean of all word embeddings in the language model vocabulary was chosen for a similar reason as the random initialisation, with the extra reasons being that it is very explicitly average and there is no chance of "bad luck" resulting in a suboptimal random initialisation. We find that this initialisation strategy performs best for both GLIP and the MM-Grounding DINO models.

- **Special Characters** such as parentheses, exclamation points, etc.: Neither of the previous initialisation strategies work well for LLMDet, which we hypothesise to be caused by language-language interactions throwing the model off when the attention sinks are too far removed from what has been seen during the LLM-aided training. Special characters, specifically brackets and parentheses, can often be found in a sentence, but do not relate to any object in particular. We find that for LLMDet, "[()]" performs best as an attention sink initialisation (which is one sink consisting of 4 tokens).

## A.3  ADDITIONAL EXPERIMENTS

**Are background false positives more common for common classes?**   To explore whether common classes drive the effect of background false positives, we split the LVIS dataset into 57 COCO-overlapping classes (our proxy for "common") and the remaining 1,146 classes (our "long tail"). Surprisingly, the results are nearly identical: both groups average 2.3 false positives per class per image, with standard deviations of 0.55 (LVIS) and 0.81 (COCO). This, combined with the observation that even entire negative phrases can trigger false positives (Figure 1), suggests that the specific content of the negative prompt does not matter.

Instead, we find a different pattern. False positives per negative prompt show a moderate positive correlation (Pearson $r = 0.34$) with the number of medium-to-large objects in the image (annotation area $\geq 64^2$). Because LVIS annotations are non-exhaustive, there are usually more large objects present than labelled, so the observed correlation may be an underestimate of the true relationship. This points to a likely explanation: the model learns to propose class-agnostic bounding boxes, which are then randomly assigned negative classes, since the corresponding features contain negative class information (Figure 4).

**Results on POPE**  Table 2 shows the results of early-fusion models, without and with attention sinks, on the POPE benchmark. All models again see large AP gains at low $FPR_{bg}$, except for Grounding DINO. This is because Grounding DINO has been trained on COCO, which has allowed it to properly calibrate its predictions on this specific combination of images and classes. However, as seen in Table 1, Grounding DINO shows significant degradation on LVIS at low $FPR_{bg}$, even though the images of both benchmarks are identical. This further shows the impracticality of handling background false positives through training on explicit negatives alone, it does not generalise to other classes without fixing the vision-language fusion layers.

| Model | $AP^{FPR}_{0.01}$ | $AP^{FPR}_{0.05}$ | $AP^{FPR}_{0.1}$ | $AP^{FPR}_{0.25}$ | AP |
|---|---|---|---|---|---|
| *Early-Fusion* | | | | | |
| GLIP-T | 0.092 | 0.302 | 0.369 | 0.436 | 0.507 |
| GDINO-T | 0.293 | 0.413 | 0.457 | 0.505 | 0.546 |
| MM-GDINO-T | 0.107 | 0.262 | 0.337 | 0.432 | 0.552 |
| LLMDet-T | 0.177 | 0.278 | 0.349 | 0.432 | 0.59 |
| *Early-Fusion + attn sinks* | | | | | |
| GLIP-T | 0.178 (+0.086) | 0.35 (+0.048) | 0.407 (+0.038) | 0.458 (+0.022) | 0.506 (-0.001) |
| GDINO-T | 0.308 (+0.015) | 0.413 (+0.0) | 0.463 (+0.006) | 0.5 (-0.005) | 0.54 (-0.006) |
| MM-GDINO-T | 0.179 (+0.072) | 0.347 (+0.085) | 0.419 (+0.082) | 0.476 (+0.044) | 0.523 (-0.029) |
| LLMDet-T | 0.202 (+0.025) | 0.417 (+0.139) | 0.458 (+0.109) | 0.488 (+0.056) | 0.519 (-0.071) |

Table 2: Results on the POPE hallucination detection benchmark, adapted to object detection, at five levels of $FPR_{bg}$. Results for Grounding DINO are indicated in gray because the model has been trained on COCO, which is the source of POPE's images and annotations. This does further show that training can mitigate background false positives for specific datasets, but does not scale to other datasets (indicated by significantly worse results of GDINO on LVIS, even though the images are the same as POPE).

## A.4 OTHER ATTEMPTS

In this section, we briefly touch on several experiments that did not perform as well as the methods in the main text, but might nonetheless be interesting for future work on this topic.

**Training attention sinks**  For all models evaluated with attention sinks in the main text, we have additionally tried training the attention sinks through textual inversion (Ruis et al., 2025), a form of prompt tuning where we only update the attention sink tokens and keep the rest of the model weights fixed. We used the VOC dataset (Everingham et al., 2010) as training data and generated background-only negatives as described in Section 5. We additionally experimented with a loss in which we explicitly mark the attention sinks as the background class. We find that this does not perform better, and in some cases even worse, compared to a training-free random or mean initialisation of the attention sinks.

**Disabling problematic attention heads**  It is possible to identify the attention heads that contribute the most to false positive predictions, e.g. by determining which attention heads have the largest activations on negative classes. We experimented with disabling or smoothing activations for such attention heads. Although this did result in performance gains, it is much less effective than attention sinks, and has a relatively large impact on positive predictions as well.

**Test-time augmentations** An interesting way to mitigate false positives is to utilise the fact that the presence of a true positive will help in the calibration of model predictions. One can copy a visual example of an image of one or more objects present in the prompt to the margins of the input image, e.g. an image of a tractor when attempting to detect tractors (cf. Figure 9). This effectively turns the background-only image into an image containing at least one true positive example, and predictions on the example image can be thrown away before evaluating. While we find that this is an effective approach for prominent objects, it fails to mitigate false positives on smaller or more difficult to see objects. It also comes with the limitations that one needs to know a-priori which objects can be prompted for and have a relevant example image ready, diminishing the open-vocabulary uses of the model, and it increases the input resolution of the image.

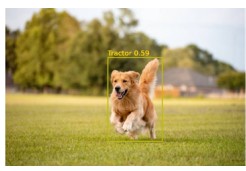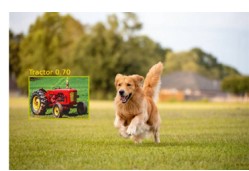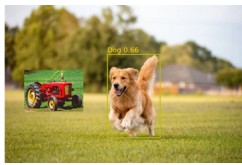

Figure 9: An example of a test time augmentation (TTA) that can effectively mitigate the false positive prediction "tractor" without affecting the true positive "dog". Placing the example in the margin outside of the image would alleviate the need to find an empty spot, and allows for cropping the image back to the original resolution while throwing away predictions outside of the image to filter TTA predictions out of the results.

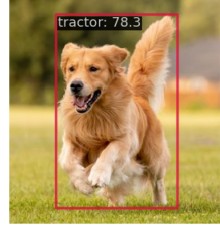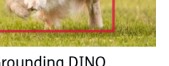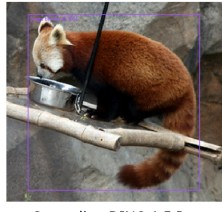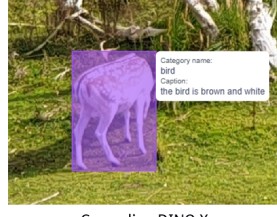

Grounding DINO
(tractor)

Grounding DINO 1.5 Pro
(jenga blocks)

Grounding DINO X
(bird)

Figure 10: Examples of confident false positives for Grounding DINO models, including the api-only models 1.5 Pro and X. Despite significantly increasing the proportion of negative samples during pre-training, 1.5 Pro and X still suffer from background-only false positives.

