# OpenReview forum: "Fantastic Tractor-Dogs and How Not to Find Them With Open-Vocabulary Detectors"
_ICLR.cc/2026/Conference — ICLR 2026 Poster_

### Official Review · Reviewer_tkvq · 2025-10-28

**Soundness:** 3
**Presentation:** 3
**Contribution:** 3
**Rating:** 6
**Confidence:** 3

**Summary:**

This paper highlights a critical limitation in Open-Vocabulary Object Detectors (OVDs): they often generate confident false positives on images that contain no relevant objects (for example, detecting a tractor in a photo of a dog).
This issue is overlooked by standard benchmarks such as COCO and LVIS, which rarely include background-only images.
The authors show that early-fusion models (e.g., GLIP, Grounding DINO) are particularly vulnerable to this problem due to information leakage in cross-modal attention, whereas late-interaction models (e.g., CLIP, OWL-ViT) are more robust.
To address this, the paper introduces a simple, training-free method that appends attention sink tokens to prompts, effectively reducing background activations and false detections without additional training.
Experimental results demonstrate consistent improvements across multiple OVD architectures.

**Strengths:**

- Clear identification of the background false positive problem ($\text{FPR}_\text{bg}$) in early-fusion OVD models.

- Strong quantitative analysis using LVIS negative labels and adapted POPE benchmark.

- Simple, training-free solution (attention sink tokens) effectively reduces $\text{FPR}_\text{bg}$ and restores $\text{AP}$.

- Consistent improvements across multiple models with minimal implementation cost.

**Weaknesses:**

- Ablation limited to initialization and count; lacks analysis of token placement or interaction.
- No runtime or efficiency analysis for large-scale inference.
- Explanation mostly empirical without theoretical grounding.

**Questions:**

- Do attention sinks affect true positive recall when target classes are present?
- What is the computational overhead in real deployment?
- Can the method generalize to domain-shifted or web-scale datasets?

---

> ### Author Response · Authors · 2025-11-21
> **response to reviewer**
>
> > Ablation limited to initialization and count; lacks analysis of token placement or interaction.
>
> We will add an improved discussion section on our earlier explorations, which should serve as a form of ablation on the nature and interaction of the target prompt with vision tokens in the absence of true positives. This includes an exploration of the disabling of specific problematic attention heads, the use of vision sinks instead of or in addition to textual sinks, and a learned explicit background class.
>
> > No runtime or efficiency analysis for large-scale inference.
> > What is the computational overhead in real deployment?
>
> This is a good question. There is no significant difference for any of the evaluated models between the inference time on, e.g., one image of VOC with 20 classes vs one image of COCO with 81 classes. Because this difference is 61 classes, i.e. > 24 tokens, the overhead by our method is negligible, because we only add a maximum of 24 tokens. Because we only add additional tokens to the class prompt, and no other architectural changes, the efficiency analysis included in the original papers of the base models still holds. We will add this discussion to the paper.
>
> > Explanation mostly empirical without theoretical grounding.
>
> To improve on the theoretical grounding, we will add a new figure about the methodology, and a discussion on the nature of the softmax attention operation and its role in spreading spurious information:
> [Figure Link](https://i.postimg.cc/HxdzLPGh/sinks-diagram.png) (hosted on postimg.cc)
>
> > Do attention sinks affect true positive recall when target classes are present?
>
> In some cases, true positive recall can be negatively impacted, an example of which we show in Figure 5. Due to limited space we initially opted to keep this analysis at a global level with AP when not accounting for background false positives (Table 1, right-most column), where for half of the models there is a slight decrease in performance and for the other half a slight increase. We will incorporate a more detailed analysis in the paper.
>
> > Can the method generalize to domain-shifted or web-scale datasets?
>
> The approach should not perform differently under domain shifts or real-world settings, at least not more than the base model itself; where all the studied models are pretrained on large, broad datasets, i.e., the models are expected to be quite robust to domain shifts and samples from web scale data. For our methodology specifically, we find that the failure mode we discovered is not dataset specific or domain dependent, but rather a model engineering issue (attention layers forcing the model to incorporate spurious label information). We will include evaluations on additional datasets such as ODinW in the appendix, to show that attention sinks will not significantly affect the base model's performance.

---

### Official Review · Reviewer_h5nZ · 2025-10-30

**Soundness:** 4
**Presentation:** 4
**Contribution:** 3
**Rating:** 6
**Confidence:** 3

**Summary:**

This paper investigates a critical issue in OVD—the tendency of early-fusion multimodal models to produce high-confidence false positives on background images (termed background hallucinations). The authors show that conventional benchmarks like COCO and LVIS rarely include pure background images, thereby masking this pathology. Through a comparative study across early-fusion (e.g., GLIP, Grounding DINO, LLMDet) and late-interaction (e.g., OWL-ViT, DetCLIP) architectures, the paper finds that early-fusion models exhibit strikingly higher false-positive rates when presented with images containing no objects of interest. To mitigate this, the authors propose a training-free “attention sink” mechanism: augmenting the text prompt with semantically neutral sink tokens that attract residual attention and absorb irrelevant activations. Extensive experiments across six state-of-the-art OVD models demonstrate that this simple modification drastically reduces background false positives (e.g., up to 5× AP improvement at FPR=0.01 on LVIS). The proposed method requires no retraining, minimal code changes, and is compatible with existing model checkpoints.

**Strengths:**

- Novel perspective: The paper introduces “background-only” testing to reveal systematic bias in OVD robustness.

- Effective and simple method: The attention sink approach is easy to implement and achieves strong performance improvements.

- Strong empirical validation: Results across six early-fusion models and multiple datasets show good generality.

- Reproducible and transparent: The authors provide code, metrics, and data splits for easy verification.

**Weaknesses:**

- Lack of theoretical grounding: The paper provides no formal model or proof explaining how attention mass redistribution through sink tokens reduces false positives.

- Missing comparison with alternative methods: Only architectural baselines are considered; other hallucination mitigation strategies are not evaluated.

- Limited dataset diversity: Experiments rely on curated datasets, leaving uncertainty about performance under domain shifts or in real-world, unlabeled settings.

- Model-specific tuning required: The optimal number, initialization, and placement of sink tokens vary across models, making the approach not fully plug-and-play.

**Questions:**

Have the authors considered hybrid approaches combining sink tokens with confidence calibration or background prompts?

---

> ### Author Response · Authors · 2025-11-21
> **response to reviewer**
>
> > Lack of theoretical grounding: The paper provides no formal model or proof explaining how attention mass redistribution through sink tokens reduces false positives.
>
> To improve on the theoretical grounding, we will add a new figure about the methodology, and a discussion on the nature of the softmax attention operation and its role in spreading spurious information:
> [Figure Link](https://i.postimg.cc/HxdzLPGh/sinks-diagram.png) (hosted on postimg.cc)
>
> > Missing comparison with alternative methods: Only architectural baselines are considered; other hallucination mitigation strategies are not evaluated.
>
> As far as we are aware there are no other existing hallucination mitigation strategies for open-vocabulary detection which we can compare to, but we are open to any suggestions of alternative methods we may have missed.
>
> > Limited dataset diversity: Experiments rely on curated datasets, leaving uncertainty about performance under domain shifts or in real-world, unlabeled settings.
>
> The approach should not perform differently under domain shifts or real-world settings, at least not more than the base model itself, because the failure mode we describe is not dataset or input dependent, but a model engineering issue (attention layers forcing the model to incorporate spurious label information). We will include evaluations on additional datasets such as ODinW in the appendix, to show more convincingly that attention sinks will not significantly affect the base model's performance.
>
> > Model-specific tuning required: The optimal number, initialization, and placement of sink tokens vary across models, making the approach not fully plug-and-play.
>
> That is indeed true, which is why we performed a search on 64 VOC images as described in the section at line 396 in the paper to find the best-performing initializations for each model. While this is quite fast and easy to perform, we do note this as a main limitation of our current approach.
>
> > Have the authors considered hybrid approaches combining sink tokens with confidence calibration or background prompts?
>
> We discuss a few other approaches in the appendix, such as test time augmentations, vision sinks, or disabling certain attention heads that play a large role in false positive predictions. We have also attempted confidence calibration, but that did not yield positive results because the false positives and their confidences are more correlated with the number of objects (labeled or otherwise) visible in the image rather than the type of object or anything that can be gleaned from its visual features (we explore this in Appendix A.3). In the end, attention sinks on their own performed very well, and additional attempts to further improve results mostly further complicated the approach with statistically non-significant gains. We will add this discussion to the paper.

---

### Official Review · Reviewer_dGwH · 2025-10-30

**Soundness:** 2
**Presentation:** 2
**Contribution:** 2
**Rating:** 4
**Confidence:** 4

**Summary:**

The paper studies zero-shot open-vocabulary detection results of foundational detectors when fed with counterfactual prompts, i.e., the prompt contain objects that do not exist in the test image. The paper finds that early-fusion detectors (i.e., fusing features of visual and textual input) output incorrect detections in front of counterfactual prompts, while late-fusion detectors are quite robust. The paper argues that early-fusion detectors are still useful in other tasks such as VQA and hence focus on early-fusion detectors including GroundingDINO and LLMDet. The paper refers to the literature and proposes to add non-learned sink tokens to improve the early-fusion detectors. The paper uses popular datasets such as COCO and LVIS to validate the proposed approach.

**Strengths:**

Below are notable strengths of this paper.
- The paper studies foundational detectors' performance with counterfactual prompts. This is an interesting topic.
- The solution by adding non-learned sink tokens is simple.
- The choice of using negative annotation in LVIS images in experiments is interesting.

**Weaknesses:**

Below are notable weaknesses of this paper.

- The motivational example in Figure 1 is not convincing. The reviewer took this image from Figure 2 and tested GroundingDINO: when using the "tractor" prompt, GroundingDINO actually did not output anything, which should be deemed a good thing. Yet, Figure 1 shows a rather high-confidence detection (78.5) on "tractor", contradicting what the reviewer observed. The paper is expected to provide more examples. Further, the paper should provide more details; for example, the rsolution and aspect ratio of the three images are different in Figure 1. Do these factors affect the detection results?

- Although the paper does not mention the word "counterfactual", there are previous works on counterfactual textual grounding, counterfactual VQA, counterfactual visual reasoning for segmentation, etc. These works share a common point as the reviewed paper that they all analyze models' predictions for counterfactual input. However, the paper does not note these related works.

- The solution by adding sink tokens are not clear. One reason is that the paper does not provide important details to understand how it works and why it can work. A diagram can help a lot. Another reason is that the paper seems to apply the sink token technique which has published by previous works. This makes the technical contribution limited.

- The visual demonstrations are quite limited. Figure 1 only provides one motivational example with three images, which are expected to be the same but actually differ in resolution and aspect ratio. Figure 5 does not show texts clearly. Figure 3 is confusing -- isn't it a good thing to obtain high LVIS MiniVal AP with decreased AP FPR (with fixed threshold 0.05)?

- The code snippet is hard to read. It is not pseudo code. It contains too much coding details and notations without comments.

**Questions:**

The reviewer asks the authors to address each point in weaknesses listed above and does not repeat them in this Questions box.

---

> ### Author Response · Authors · 2025-11-21
> **response to reviewer, part 1**
>
> > The motivational example in Figure 1 is not convincing. The reviewer took this image from Figure 2 and tested GroundingDINO: when using the "tractor" prompt, GroundingDINO actually did not output anything, which should be deemed a good thing. Yet, Figure 1 shows a rather high-confidence detection (78.5) on "tractor", contradicting what the reviewer observed. The paper is expected to provide more examples. Further, the paper should provide more details; for example, the rsolution and aspect ratio of the three images are different in Figure 1. Do these factors affect the detection results?
>
> The reviewer is right that small image changes will affect the model’s predictions. This may lead to different box/label predictions for the same image under small variations, as observed by the reviewer on Figure 1. Figure 1 is a true observation on an image from the web when the Grounding DINO Swin-T model is applied out of the box. It is a larger image which we have cropped to only include predicted bounding boxes, as the full image would take up too much space.
>
> We agree with the reviewer that a single illustration is never satisfactory or convincing on its own, Figure 1 mainly serves as a hopefully intuitive example of the failure case we discovered. To demonstrate that Figure 1 is just one example of an undesired phenomenon that happens structurally for a wide range of early-fusion models, we quantify this problem in a set of large-scale experiments on multiple datasets and models.
>
> > Although the paper does not mention the word "counterfactual", there are previous works on counterfactual textual grounding, counterfactual VQA, counterfactual visual reasoning for segmentation, etc. These works share a common point as the reviewed paper that they all analyze models' predictions for counterfactual input. However, the paper does not note these related works.
>
> This is a good point by the reviewer. Counterfactual analysis can indeed be an alternative approach to reduce false positives. We agree that this is an interesting angle to the problem, but it requires tailored training schemes, as for counterfactual methods the input is deliberately modified to create “what if” scenarios, e.g. removing, replacing, or altering objects or attributes in an image to analyze how a model’s prediction changes.
>
> Instead, our starting point was to propose a training-free approach, which is attractive, because it is simpler and requires no training time or training compute resources, and still very effective, as shown by our experiments. Our analysis therefore focuses on the model’s intrinsic bias and spurious activations in the absence of relevant visual evidence, not on causal reasoning under counterfactual manipulations.
>
> > The solution by adding sink tokens are not clear. One reason is that the paper does not provide important details to understand how it works and why it can work. A diagram can help a lot.
>
> The essence is that in the vision-language fusion layer, e.g. the Feature Enhancer in Grounding DINO, image and text features are fused through cross attention. The attention scoring operation cannot output 0, so the fusion layer will always have to include information of at least one of the target classes in any one of the image patch features. Those same text features are then later used to classify object queries based on how much textual information that query contains.
>
> After the fusion, the language-guided query selection utilizes pre-trained query embeddings to aggregate information about objects in the image. Due to the open-set nature of the detector these queries are class-agnostic and are able find most objects in the provided image. In our example above, one query would e.g. aggregate all visual information of the dog (including the fused textual information) into one query embedding.
>
> Finally, the queries obtain a classification score based on how much information about each of the target classes is included in the query. During training, the model has learned to calibrate its confidence such that one of the target classes gains a high score. This is not an issue when a clear positive class is present (e.g. the dog, tractor example), but this becomes problematic when no positive class is present (e.g. the tractor, pizza example).
>
> By adding attention sinks, which are specifically chosen to have a high likelihood to be picked by the vision-language fusion layer in the absence of a positive class, we can control this flow of information and can then discard any queries that have a high activation score with a sink token.
>
>
> This is shown in the diagram below, which will be added to the paper, together with the explanation:
> [Figure Link](https://i.postimg.cc/HxdzLPGh/sinks-diagram.png) (hosted on postimg.cc)

---

> > ### Author Response · Authors · 2025-11-21
> > **response to reviewer, part 2**
> >
> > > Another reason is that the paper seems to apply the sink token technique which has published by previous works. This makes the technical contribution limited.
> >
> > Indeed, there are several papers on attention sinks or similar, e.g. Vision Transformers Need Registers (Darcet et al.) and more recently, Kang et al. (https://arxiv.org/abs/2503.03321). To our knowledge, no paper on sink tokens has addressed the object detection task. Notably, our technical contribution is mostly the discovery of the problem of false positives in backgrounds in (early fusion) open-vocabulary object detection models and the finding that it can be improved significantly with a simple recipe that incorporates attention sinks into such models, where the attention sinks are mostly a tool to achieve that goal.
> >
> > > The visual demonstrations are quite limited. Figure 1 only provides one motivational example with three images, which are expected to be the same but actually differ in resolution and aspect ratio. Figure 5 does not show texts clearly.
> >
> > We agree, we will include additional qualitative examples and improve the readability of figure 5.
> >
> > > Figure 3 is confusing -- isn't it a good thing to obtain high LVIS MiniVal AP with decreased AP FPR (with fixed threshold 0.05)?
> >
> > We now understand that it is confusing indeed. Figure 3 is not a conventional AP vs. FPR plot. Instead, it is an AP vs AP’ plot, where AP’ is the AP at a fixed (low) FPR. The goal of the figure is to show that standard AP (i.e. unbounded FPR) and AP’ (i.e. low FPR) are very different, while, ideally, they should have the same value (x=y) if there are not many false positives on background images (y-axis). Hence, ideally the plotted points should be close to the diagonal (x ~ y). But they are not, typically y << x. This exemplifies that many models suffer from high FPR on background images, and that this leads to lower AP when thresholding on FPR (y << x). This is the finding that we want to emphasize here, and that this can be solved by our methodology (which improves on the y values across the models).
> >
> > For example, LLMDet Swin-B without attention sinks reaches an AP of 46.6 with unbounded FPR (x-axis), but only 12.5 with FPR fixed at 0.05 (y-axis), i.e., its performance drops by 34.1.
> >
> > >The code snippet is hard to read. It is not pseudo code. It contains too much coding details and notations without comments.
> >
> > We agree, we will improve the paper by adding pseudo code to understand the main idea first, and then list the actual code to show the implementation of the idea, and refer to them with the right terms.

---

> > > ### Comment · Reviewer_dGwH · 2025-11-28
> > >
> > > The reviewer appreciates the rebuttal, which addresses some of my concerns. For example, the explanation and diagram of sink tokens help a lot. The paper should incorporate them. However, the reviewer is not convinced by the limited visual demonstrations and the lack of discussions with closely related works including counterfactual grounding and open-set/open-world object detection (as the authors mentioned "open-set nature" in the rebuttal). Given the existing areas of counterfactual grounding and open-set/open-world object detection, it is not surprising that open-vocabulary detectors occasionally fire in face of non-existing or counterfactual objects in prompts. Moreover, unlike the rebuttal which states "Counterfactual analysis can indeed be an **alternative approach**", the reviewer would compare the related works from perspectives of motivations and empirical observations, rather than methodology. At this point, the paper is expected to discuss the related literature to better show its insights, position and significance.

---

> > > > ### Author Response · Authors · 2025-12-03
> > > >
> > > > We respectfully disagree on counterfactual grounding being closely related to our work. This may stem from different interpretations of the term counterfactual, but we do not make use of any counterfactual prompts or images, as there is no modification to a factual prior event. The negative samples are and always were simply background images, just like background images in traditional detection approaches. All related work we can find on counterfactual grounding is related to causality and/or explainability, we are unable to find approaches that could be considered similar or in any way related to ours, even when expanding our search to other areas than object detection. As such, we feel it would be confusing to most readers to include it in our related work discussions.
> > > >
> > > > We would like to re-iterate that, while a counterfactual approach may help in finding out why certain mistakes are made by a model, that is mostly relevant for reasoning or feature learning errors, such as the remaining false positives we describe in Section 6 (e.g. a regular pumpkin detected as a jack-o’-lantern). However, that is not the type of issue we are tackling in this paper. We instead identify a __model engineering__ error that is unrelated to the model's training or reasoning.

---

### Official Review · Reviewer_9wAb · 2025-11-04

**Soundness:** 3
**Presentation:** 4
**Contribution:** 2
**Rating:** 8
**Confidence:** 3

**Summary:**

This paper shows a flaw in real-world scenarios for OVD, frequently make high-confidence false positive predictions on images that do not contain the target object. For instance, prompting an OVD for "tractor" on an image of a dog often results in a confident "tractor" detection. This issue is masked by standard benchmarks like COCO and LVIS, which rarely contain images completely lacking the target classes.The authors identify that this issue specifically affects early-fusion architectures, where vision and language features are combined early via cross-attention. Late-interaction models (e.g., CLIP) do not suffer this significant deviation in performance on background images. To mitigate this without costly retraining, the authors propose a simple, training-free method, appending "attention sink" tokens to the input prompt to act as "none of the above" type tokens. The authors evaluated this method on six early-fusion models using modified LVIS and POPE benchmarks.

**Strengths:**

The paper identifies a major issue in real-world deployments of Open-Vocabulary Detectors (OVDs), the false positives were previous unknown.. It also explains why this issue has been missed, noting that standard benchmarks like COCO and LVIS don't show the problem because the images rarely don't contain the target.

The analysis is good. They demonstrate that these layers cannot "pick zero tokens" when a match is absent, forcing them to attend to irrelevant tokens. They also clearly differentiate this behavior from late-interaction models, which do not suffer from this significant deviation on background images.

The proposed solution, appending attention sink tokens to the prompt is nice, and training-free and works out-of-the-box for all evaluated models. The implementation is highlighted as very straightforward, requiring only a few lines of code.

The attention sink approach reduces background false positives, and the results are strong.

The paper proposes simple adaptations to existing benchmarks to quantify this issue, such as leveraging negative labels in LVIS to calculate a specific background false positive rate.

**Weaknesses:**

1. Limited Exploration of "Attention Sink" Design Space and Generalizability. While the proposed "attention sink" method is elegant and effective, the paper's exploration of its design space is limited, there could be more options explored and ablations done. However, it does work, so this is not a major weakness.

2. The authors use random initializations, mean embeddings, and special characters. Do these different types of sinks perform equally across different model architectures (e.g., are Grounding DINO's cross-attention layers equally receptive to a random vector vs. a mean embedding)?

3. The paper does not thoroughly explore the impact of the number of sinks. Is one sink token sufficient, or is there a diminishing return after a certain number? This is critical for practical deployment, as more tokens increase inference time.

4. How does the token length of the class prompt (e.g., "a single dog" vs. "a fantastic black-and-white tractor-dog running across a field") interact with the optimal number or type of sink tokens?

5. The core goal is to reduce False Positives (e.g., detecting a tractor on a dog), which is a True Negative case (no tractor is present). While the results show gains in AP/FPR on background images, the paper lacks a detailed analysis of potential collateral damage to True Positives (detecting the dog when the dog is actually present).

6. The mechanism works by providing an alternative to irrelevant class tokens. A key concern is whether a confident true detection, where the vision features were already weak, is accidentally routed to the sink instead of the correct class. A sensitivity analysis showing how the confidence of marginal correct detections changes would strengthen the conclusion.

7. The paper claims a training-free solution, but the conceptual problem of "null-class" or "background" modeling is not new in object detection. For instance, how does the sink mechanism relate to the concept of a "background class" token used in traditional detectors (like a final output class)? While the OVD context is different, clarifying the relationship to this established concept would enhance the paper's grounding in detection literature.

8. The paper states that late-interaction models (like CLIP-based methods) do not exhibit the same catastrophic failure mode. However, this conclusion is based on a limited scope.

**Questions:**

See weaknesses.

---

> ### Author Response · Authors · 2025-11-21
> **response to reviewer, part 1**
>
> > Limited Exploration of "Attention Sink" Design Space and Generalizability. While the proposed "attention sink" method is elegant and effective, the paper's exploration of its design space is limited, there could be more options explored and ablations done. However, it does work, so this is not a major weakness.
>
> We agree and will add a discussion on the design space and generalizability of the attention sinks, where we will include more of our earlier explorations. A logical alternative to our textual sinks in the prompt is to use vision sinks, which we briefly describe in Section 4. Another alternative to sinks we explored, is to adapt the attention heads, where spurious information may also be redirected or switched off. Our explorations on the attention heads resulted in marginal improvements, while also being ad-hoc in terms of strategy, leading to non-trivial solutions that did not generalize well. In the end, we opted to go for simplicity, which is why we are grateful to hear you describe the method as elegant.
>
> > The authors use random initializations, mean embeddings, and special characters. Do these different types of sinks perform equally across different model architectures (e.g., are Grounding DINO's cross-attention layers equally receptive to a random vector vs. a mean embedding)?
>
> This is a good point. Unfortunately, they do not. Ideally, one good strategy would fit all models, but that is not the case. Therefore, we were curious how little data is needed to select the right strategy for the current specific model at hand. The interesting finding is that as few as 64 images from VOC are sufficient to select the best strategy for the given model. This finding is shown on line 396 in the paper. We will add some text to exemplify the rationale for this experiment in terms of the raised items by the reviewer.
>
> > The paper does not thoroughly explore the impact of the number of sinks. Is one sink token sufficient, or is there a diminishing return after a certain number? This is critical for practical deployment, as more tokens increase inference time.
>
> Indeed, this is an important question. We describe this on page 8, and Figure 7 (right). As hinted by the reviewer, a quite limited number of tokens are sufficient, after which the performance gains are reaching a plateau. Among all models evaluated, 24 sinks is the maximum number of sinks to reach optimal performance, with most models performing quite well at around 4 to 8 sinks. Adding more tokens is not effective, but also not harmful. We will update the manuscript to include more details on the optimal number of sinks for each of the models in the appendix.
>
> > How does the token length of the class prompt (e.g., "a single dog" vs. "a fantastic black-and-white tractor-dog running across a field") interact with the optimal number or type of sink tokens?
>
> This is a good question. Surprisingly, we observed no differences to the optimal number or type of sink tokens based on class prompt length (such as target phrases in referring expression comprehension) or number of classes. We will mention this in the revised paper.
>
> > The core goal is to reduce False Positives (e.g., detecting a tractor on a dog), which is a True Negative case (no tractor is present). While the results show gains in AP/FPR on background images, the paper lacks a detailed analysis of potential collateral damage to True Positives (detecting the dog when the dog is actually present).
> The mechanism works by providing an alternative to irrelevant class tokens. A key concern is whether a confident true detection, where the vision features were already weak, is accidentally routed to the sink instead of the correct class. A sensitivity analysis showing how the confidence of marginal correct detections changes would strengthen the conclusion.
>
> This does indeed happen in some cases, an example of which we show in Figure 5. Due to limited space we initially opted to keep this analysis at a global level with AP when not accounting for background false positives (Table 1, right-most column), where for half of the models there is a slight decrease in performance and for the other half a slight increase. We will incorporate a more detailed analysis in the paper.

---

> > ### Author Response · Authors · 2025-11-21
> > **response to reviewer, part 2**
> >
> > > The paper claims a training-free solution, but the conceptual problem of "null-class" or "background" modeling is not new in object detection. For instance, how does the sink mechanism relate to the concept of a "background class" token used in traditional detectors (like a final output class)? While the OVD context is different, clarifying the relationship to this established concept would enhance the paper's grounding in detection literature.
> >
> > This is a good point. In our initial explorations to solve this problem we have toyed with bringing back a more explicit background class (indeed inspired by traditional detectors), but that did not generalize well outside of the set of classes it was trained on, and risks catastrophic forgetting. The sinks are not directly related to a background class, the sinks are only there to redirect spurious attention in the vision-language fusion layers. We find that the model is actually quite good at distinguishing between the target classes and background if it has a way to discard irrelevant information, which is what the sinks provide. We will add a discussion about this alternative strategy, with some references to established literature, and its limited effectiveness in the light of OVD background false positives.
> >
> > > The paper states that late-interaction models (like CLIP-based methods) do not exhibit the same catastrophic failure mode. However, this conclusion is based on a limited scope.
> >
> > The reviewer is right, we will add to the revised paper that this can only be concluded for OVD settings, i.e. object detection, where we observe and quantify an interesting difference between early-fusion and late-interaction models, the first category exhibiting more problems regarding false positives on backgrounds.

---

### Meta-Review · Area_Chair_Apgc · 2026-01-04

**Summary:**

The paper identifies and analyzes the problem of false positives in open-vocabulary detectors - e.g., confidently detecting a tractor in an image of a dog, when no alternative positive class is requested. After the analysis, a method is proposed to fix it - "attention sinks", additional cross-attention tokens that can be used by the model to route irrelevant information. The method is evaluated on several benchmarks and demonstrates large improvements in terms of false positives compared to the original detectors.

The reviewers are mostly positive about the paper, with one exception. Here are key mentioned pros and cons.

Pros:
1. Identifying and highlighting an important problem with open-vocabulary detectors.
2. Thorough quantitative analysis of the problem.
3. A simple and effective solution via "attention sinks"

Cons:
1. No theoretical justification.
2. Limited comparison to alternative methods, including counterfactual analysis.
3. Few qualitative examples.

The rebuttal aimed to address the issues. It claimed alternatives methods are not available and conterfactual methods are of limited relevance. I believe theoretical justification is not crucially reuqired as long as the method works very well empirically. More qualitative examples would be great indeed

Overall, I concur with the majority of the reviewers and, given that the paper identifies an important problem and proposes a simple but effective solution, recommend acceptance.

I strongly encourage the authors to implement the proposed improvements incuding the schematic figure with sinks. More qualitative examples would help a lot as well.

**Reviewer Concerns:**

See the main meta-review. I think the concerns about too few qualitative examples and potentially incomplete related work section are still relevant.

**Reviewer Scores:**

The negative revier (score 4) said their concerns were partially addressed. they may have changed their schore to 5.

---

### Decision · Program_Chairs · 2026-01-26

Accept (Poster)